# Retrieval-Augmented Language Model for Knowledge-aware Protein Encoding

## Abstract

Protein language models often struggle to capture the biological functions encoded within protein sequences due to their lack of factual knowledge (*e.g.,* gene descriptions of proteins). Existing solutions leverage protein knowledge graphs (PKGs), using knowledge as auxiliary encoding objectives. However, none of them explored the direct injection of correlated knowledge into protein language models, and task-oriented knowledge integration during fine-tuning, making them suffer from insufficient knowledge exploitation and catastrophic forgetting of pre-trained knowledge. The root cause is that they fail to align PKGs with downstream tasks, forcing their knowledge modeling to adapt to the knowledge-isolated nature of these tasks. To tackle these limitations, we propose a novel knowledge retriever that can accurately predict gene descriptions for new proteins in downstream tasks and thus align them with PKGs. On this basis, we propose **K**nowledge-**a**ware **r**etrieval-**a**ugmented protein language model (**Kara**), achieving the first unified and direct integration of PKGs and protein language models. Using the knowledge retriever, both the pre-training and fine-tuning stages can incorporate knowledge through a unified modeling process, where contextualized virtual tokens enable token-level integration of high-order knowledge. Moreover, structure-based regularization is introduced to inject function similarity into protein representations, and unify the pre-training and fine-tuning optimization objectives. Experimental results show that Kara consistently outperforms existing knowledge-enhanced models in 6 representative tasks, achieving on average 5.1% improvements.

## 1 Introduction

Proteins are essential for understanding biological processes and recent advances in artificial intelligence led to growing interest in learning generalized vector representations of proteins (Hu et al., 2024). By viewing amino acids as language tokens, protein language models (PLMs) such as ESM (Lin et al., 2023), ProteinBert (Brandes et al., 2022), and ProtBert (Ahmed et al., 2020) have proven highly valuable in various application tasks such as drug discovery (Hoang et al., 2024) and function prediction (Xu et al., 2024; Shaw et al., 2024). However, as pointed out by Kalifa et al. (2024); Zhou et al. (2023); Zhang et al. (2022), lacking factual knowledge (*e.g.,* gene descriptions) makes them struggle to capture intricate biological function encoded within protein sequences.

Existing solutions leverage protein knowledge graphs (PKGs) that describe the relationships between proteins and gene ontology (GO) entities with biological relations (Chen et al., 2023b). These models use protein sequences and associated GO annotations as complementary encoding objectives to infuse knowledge information. For example, OntoProtein (Zhang et al., 2022) uses knowledge embedding objective (*i.e.,* TransE (Bordes et al., 2013)) to optimize the alignment between the protein representations and associated GO entity representations. KeAP (Zhou et al., 2023) integrates GO entity representations into the masked token prediction of protein sequences through a cross-attention mechanism. Despite their effectiveness, unfortunately, they still have several limitations.

**Limitations. 1) Implicitly embed knowledge information.** Existing methods use knowledge only as encoding objectives to supervise the pre-training of the model, assuming that knowledge information can be well embedded within model parameters. However, as highlighted by Kandpal et al. (2023), LMs often struggle to precisely embed knowledge, particularly long-tail knowledge. Storing knowledge within model parameters also makes them unable to adapt to knowledge graph updates

(e.g., adding new knowledge), which further diminishes their usability. **2) Overlook the structure information.** Existing methods treat each knowledge triplet (*i.e.,* $(protein, relation, GO)$) independently. However, the neighboring GO entities of a protein are often correlated, and the high-order connections between proteins (*e.g.,* proteins linked to a GO entity through similar relations) can provide additional insights into their functional similarities. Ignoring the structural relevance makes existing methods fail to fully exploit knowledge information within PKGs. **3) Inconsistent knowledge modeling.** Existing methods incorporate knowledge modeling during pre-training but ignore it during fine-tuning, leading to inconsistent optimization objectives between these two stages. This inconsistency can cause the knowledge learned during pre-training to be catastrophically forgotten during fine-tuning (Lee et al., 2020), making it difficult to transfer to downstream tasks. Overall, the root cause of these limitations is that proteins in downstream tasks often fall outside the PKG, restraining the use of knowledge during fine-tuning. Existing methods fail to align knowledge graphs with downstream tasks, forcing their knowledge modeling to adapt to the knowledge-isolated nature of these tasks (*e.g.,* knowledge cannot be directly used as part of the input for protein encoding).

**Proposed Work.** To tackle these limitations, we propose **Kara**, a **K**nowledge-**a**ware **r**etrieval-**a**ugmented protein language model, achieving the first unified and direct integration of PKGs and protein language models. As the core of Kara, we propose a knowledge retriever that can accurately predict potential gene descriptions for new proteins and thus align them with PKGs. This alignment allows the pre-training and fine-tuning stages of Kara to be enhanced through a unified knowledge modeling process, and seamlessly adapt to knowledge updates. By employing contextualized virtual tokens, we achieve token-level information fusion between protein sequence and knowledge. Specifically, we categorized the virtual tokens into knowledge tokens and structure tokens, enabling the direct injection of both knowledge information and high-order structure information into protein representations. To unify the optimization objectives, we incorporate structure-based regularization into both two stages, injecting function similarities into protein representations and helping the pre-trained knowledge to be effectively transferred to downstream tasks.

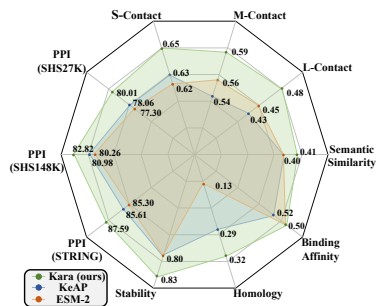

Figure 1: Performance in downstream tasks. S-, M-, and L-Contact are the short-range, medium-range, and long-range contact prediction. PPI is the protein-protein interaction prediction.

As shown in Figure 1, experiments in 6 representative downstream tasks demonstrate the effectiveness of Kara. It consistently outperforms powerful baselines (*i.e.,* KeAP and ESM-2) across all the tasks. For instance, Kara exceeds the state-of-the-art knowledge-enhanced model KeAP by $11.6\%$ in the long-range contact prediction and by $10.3\%$ in the protein homology detection, highlighting Kara as a better paradigm for integrating protein knowledge graphs into protein language models.

## 2 PRELIMINARY

**Protein Knowledge Graphs.** A protein knowledge graph (PKG) is $G = \{V_p, V_{go}, R, F\}$, where $V_p$ is the protein set and $V_{go}$ is the gene ontology (GO) entity set. $R$ is the set of relations among proteins and GO entities. The knowledge set $F$ consists of two kinds of triplets: $(p, r, g)$ which describes the properties of proteins, and $(g_1, r, g_2)$ which describes the relationships between GO entities. Each protein $p \in V_p$ has an amino acid sequence $s$. Each GO entity $g \in V_{go}$ includes a text description $t_g$ explaining the gene's function. Similarly, each relation $r \in R$ comes with a text description $t_r$. We first generate pre-trained embeddings of items in PKG and store them in vector databases for further usage. Specifically, relation $r$ and GO entity $g$ are encoded based on their text descriptions using a frozen PubMedBERT model (Gu et al., 2021), resulting in relation embedding **r** and GO embedding **g**. Protein $p$ is encoded based on its amino acid sequence via a frozen ProtBert model (Ahmed et al., 2020), resulting in protein embedding **p**. These stored embeddings will be further used to construct virtual tokens in Kara. Following previous works, we use the ProteinKG25 knowledge graph (Zhang et al., 2022). Detailed introduction of ProteinKG25 can be found in Appendix A.

**Problem Formulation.** Given a PKG $G$, we aim to pre-train a knowledge-aware protein language model $f$ so that for each protein with amino acid sequence $s$, we can generate its knowledge-

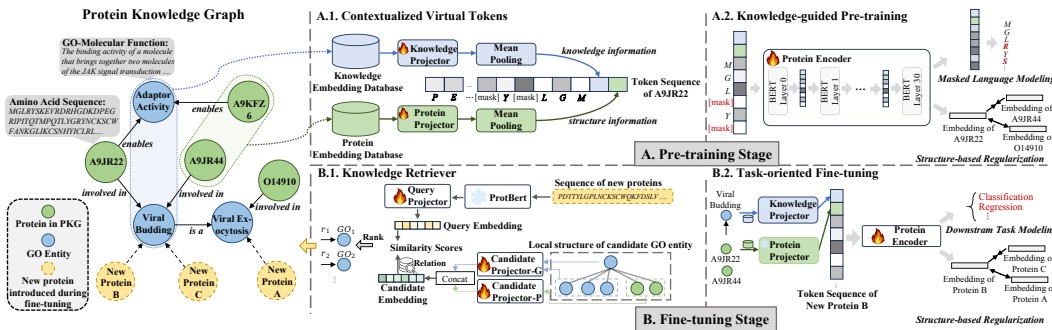

Figure 2: **Overall architecture**. During pre-training, Kara directly integrates knowledge information via contextualized virtual tokens and structure-based regularization. During fine-tuning, the knowledge information can be similarly integrated into protein representations through a knowledge retriever, which can align new proteins in downstream tasks with the protein knowledge graph.

integrated vector representation as $\tilde{\mathbf{p}} = f(G, s)$. In Kara, $f$ consists of a protein encoder, a knowledge projector, a protein projector, and a knowledge retriever. We use the ProtBert model (Ahmed et al., 2020) as the backbone of the protein encoder, which is the same as previous works (Zhou et al., 2023) for a fair comparison. By fine-tuning $f$ on task-specific data, we further verify its capabilities to generalize pre-trained knowledge to downstream tasks (*e.g.,* protein homology detection).

## 3 METHODOLOGIES

As shown in Figure 2, with a knowledge retriever to align new proteins with the protein knowledge graph, Kara can uniformly integrate knowledge information during both the pre-training and fine-tuning stages [1]. Specifically, the contextualized virtual tokens allow Kara to directly inject the associated knowledge information and high-order structure information of a protein into its representations. During pre-training, masked language modeling (MLM) helps the protein encoder learn to fuse the information of protein sequences and structured knowledge at the token level. During fine-tuning, downstream task modeling helps the protein encoder learn to extract task-specific useful knowledge from PKGs via virtual tokens. Additionally, based on the high-order connectivity between proteins, structure-based regularization is incorporated during the two stages to unify their optimization objectives and inject function similarities into protein representations. We detail each part of Kara in the following and summarize important notations used in this paper in Table 1.

### 3.1 PRE-TRAINING STAGE

#### 3.1.1 CONTEXTUALIZED VIRTUAL TOKENS

Existing protein language models struggle to encode knowledge information since 1) knowledge in PKG is interconnected, providing the context of proteins based on the graph structure, but language models are only designed to encode sequential data, limiting their ability to capture graph information; and 2) PKGs contain multi-modal information (*e.g.,* amino acid sequences and GO text descriptions), and protein language models can only encode amino acid sequences, failing to achieve effective multi-modal information fusion. As shown in Figure 2 A.1, we tackle the above challenges by introducing contextualized virtual tokens. By summarizing the associated knowledge of a protein as knowledge virtual tokens and summarizing its high-order structure as structure virtual tokens, Kara can directly inject the knowledge and graph information into protein representations. These virtual tokens are then concatenated with the amino acid token sequences as the knowledge context, so that each amino acid can query them to integrate helpful knowledge information, enabling effective token-level multi-modal information fusion. Specifically, for each protein $p_i \in V_p$, we extract its one-hop GO entities with relations $\mathcal{N}_1(p_i) = \{(r_i, g_i)|(p_i, r_i, g_i) \in F\}$ as its knowledge, and use its two-hop proteins $\mathcal{N}_2(p_i) = \{p_j|(p_j, r_i, g_i) \in F; (r_i, g_i) \in \mathcal{N}_1(p_i)\}$ as its structure

---

[1]Note that the knowledge retriever is only used during fine-tuning, ensuring that no data from downstream tasks can be leaked into the pre-training stage.

Table 1: Important notations and descriptions.

| Notation | Description |
|---|---|
| $G$ | A protein knowledge graph. |
| $V_p, V_{go}, R$ | Protein set, GO entity set, and relation set in $G$. |
| $F$ | Set of triplets (*i.e.,* knowledge) in $G$. |
| $p_i, r_j, g_k$ | A protein, a GO entity, and a relation. |
| $s_i, s_i^m$ | The amino acid sequence of protein $p_i$, and each amino acid in $s_i$. |
| $\mathbf{p}_i, \mathbf{r}_j, \mathbf{g}_k$ | Stored pre-trained embeddings of protein $p_i$, relation $r_j$, and GO entity $g_k$ (see Section 2). |
| $\mathbf{v}_i^k, \mathbf{v}_i^p$ | Knowledge virtual token and structure virtual token of protein $p_i$. |
| $\mathbf{S}_i, \mathbf{S}_i^L$ | Input embedding sequence of protein $p_i$, embedding sequence at the $L$-th layer. |
| $\tilde{\mathbf{p}}_i$ | Encoded embedding of protein $p_i$ by Kara. |
| $\mathbf{g}_m^{go}, \mathbf{g}_m^{prot}$ | Neighboring GO entity embedding and neighboring protein embedding of GO entity $g_m$. |
| $\mathbf{q}_n$ | Query embedding corresponds to new protein $p_n$. |
| $\tilde{\mathbf{r}}_m$ | Query embedding corresponds to relation $r_m$. |
| $\mathbf{c}_m$ | Candidate embedding corresponds to GO entity $g_m$. |
| $\mathbb{S}(\cdot)$ | Score function. |
| $\text{MLP}(\cdot)$ | Trainable multi-layer perceptron. |
| $N_1(p_i)$ | One-hop GO entities with relations of protein $p_i$ |
| $N_2(p_i)$ | Two-hop connected proteins of protein $p_i$. |
| $N_{go}(g_m)$ | One-hop neighboring GO entities of GO entity $g_m$. |
| $N_{prot}(g_m)$ | One-hop neighboring proteins of GO entity $g_m$. |
| $\mathcal{E}(r_m)$ | Candidate GO entity set corresponding to relation $r_m$. |

context. The knowledge virtual token of protein $p_i$ is then constructed as

$$\mathbf{v}_i^k = \frac{1}{|\mathcal{N}_1(p_i)|} \sum_{(r_i, g_i) \in \mathcal{N}_1(p_i)} \text{MLP}_{knowledge}([\mathbf{r}_i : \mathbf{g}_i]), \tag{1}$$

where $\mathbf{r}_i$ and $\mathbf{g}_i$ are respectively the pre-trained embeddings of relation $r_i$ and GO entity $g_i$ (see Section 2). $[:]$ is the concatenation operation. $\text{MLP}_{knowledge}$ is a trainable multi-layer perceptron used to project text-modal information into a uniform semantic space. Similarly, to incorporate the structure information of $p_i$, we construct its structure virtual token as

$$\mathbf{v}_i^p = \frac{1}{|\mathcal{N}_2(p_i)|} \sum_{p_j \in \mathcal{N}_2(p_i)} \text{MLP}_{structure}(\mathbf{p}_j), \tag{2}$$

where $\mathbf{p}_j$ is the pre-trained embedding of protein $p_j$. $\text{MLP}_{structure}$ is another trainable multi-layer perceptron used to project the amino acid sequence-modal information. We then construct the input embedding sequence for the protein encoder by concatenating virtual tokens with amino acid tokens. Given the amino acid sequence $s_i = [s_i^1, s_i^2, ..., s_i^{|s_i|}]$ of protein $p_i$, where $s_i^m$ represents an amino acid, we lookup the embedding vocabulary of protein encoder to initialize the input embedding sequence as $\mathbf{S}_i = [\mathbf{s}_i^1, \mathbf{s}_i^2, ..., \mathbf{s}_i^{|s_i|}] \in \mathbb{R}^{|s_i| \times d}$, then concatenate it as

$$\mathbf{S}_i \leftarrow [\mathbf{v}_i^k, \mathbf{v}_i^p, \mathbf{S}_i] \in \mathbb{R}^{(2+|s_i|) \times d}, \tag{3}$$

where $|s_i|$ is the length of amino acid sequence $s_i$, and $d$ is the dimension of embeddings. During inference, any related knowledge updates can be perceived by constructing these virtual tokens.

### 3.1.2 Knowledge-guided Pre-training

The pre-training of Kara has two purposes: 1) achieving effective information fusion of the contextualized virtual tokens (*i.e.,* knowledge and structure information) and the amino acid tokens (*i.e.,* protein information); and 2) integrating the knowledge-based relevance (*i.e.,* function similarities) among proteins into their representations. For the first purpose, we introduce knowledge-guided masked language modeling, allowing each amino acid to query the virtual tokens to extract helpful knowledge information for restoring masked tokens, which achieves token-level information fusion at each layer of the protein encoder. Specifically, given the input embedding sequence $\mathbf{S}_i$, we use a 15% probability to mask each amino acid token (*i.e.,* replace the amino acid embedding as the

embedding of special token '[MASK]'). Then the masked embedding sequence is encoded by the Transformer component (Vaswani et al., 2017) within the protein encoder as follows:

$$\tilde{\mathbf{S}}_i^l = \text{LN}(\mathbf{S}_i^l + \text{MHA}(\mathbf{S}_i^l)), \tag{4}$$

$$\mathbf{S}_i^{(l+1)} = \text{LN}(\tilde{\mathbf{S}}_i^l + \text{MLP}(\tilde{\mathbf{S}}_i^l)), \tag{5}$$

where $\mathbf{S}_i^0$ is initiated by $\mathbf{S}_i$. LN denotes the layer-norm unit and MHA denotes the multi-head attention unit. After modeling the correlations among virtual tokens and amino acid tokens layer by layer, we leverage cross-entropy loss $\mathcal{L}_{\text{MLM}}$ on the last layer token embeddings (*i.e.*, $\mathbf{S}_i^L$, where $L$ is the number of Transformer layers in protein encoder) to estimate the masked tokens.

While the aforementioned masked language modeling achieves token-level multi-modal knowledge infusion, we further introduce a sequence-level regularization based on graph connectivity between proteins, integrating biological function similarities into their representations. As we mentioned before, each protein $p_j \in \mathcal{N}_2(p_i)$ is two-hop connected with $p_i$ in graph structure. This high-order connectivity indicates that $p_i$ and $p_j$ share the same knowledge $(r_i, g_i)$ and thus should be similar in their biological functions. Therefore, each pair $(p_i, p_j \in \mathcal{N}_2(p_i))$ can be regarded as positive pair that we hope their embeddings are closer in semantic space (*e.g.,* A9JR22 and A9JR44 in Figure 2), and $(p_i, p_k \notin \mathcal{N}_2(p_i))$ can be regarded as negative pair (*e.g.,* A9JR22 and O14910). Specifically, in Kara, we generate the sequence-level embedding of protein $p_i$ as $\tilde{\mathbf{p}}_i = \text{MEAN}(\mathbf{S}_i^L[2:])$, where MEAN is the mean-pooling operation, and $\mathbf{S}_i^L[2:]$ is the last layer token embeddings except the virtual tokens. Then, we apply the margin loss on sequence-level protein embeddings to ensure high-order connected protein $p_j$ is closer to $p_i$ than other proteins in semantic space.

$$\mathcal{L}_{\text{reg}} = -\frac{1}{|\mathcal{N}_2(p_i)|} \sum_{p_j \in \mathcal{N}_2(p_i)} \text{MAX}(0, \text{sim}(\tilde{\mathbf{p}}_i, \tilde{\mathbf{p}}_j)) - \text{sim}(\tilde{\mathbf{p}}_i, \tilde{\mathbf{p}}_k) + \gamma), \tag{6}$$

where sim indicates the similarity function (*e.g.,* cosine similarity). We finally pre-train the parameters within the protein encoder, knowledge projector, and structure projector by jointly optimizing $\mathcal{L}_{\text{MLM}}$ and $\mathcal{L}_{\text{reg}}$. These three components are then used to handle downstream tasks.

## 3.2 FINE-TUNING STAGE

### 3.2.1 KNOWLEDGE RETRIEVER

Proteins in downstream tasks often fail outside the PKGs (Zhou et al., 2023), restraining the use of knowledge during fine-tuning. Existing methods thus incorporate knowledge modeling solely during pre-training, leaving the fine-tuning process only guided by task-specific objectives. However, this strategy has several limitations. 1) The optimization objectives of the pre-training and fine-tuning stages are inconsistent (*i.e.,* one is knowledge-guided while the other is knowledge-isolated), causing the pre-training knowledge to be catastrophically forgotten during fine-tuning (Lee et al., 2020). 2) Without PKGs during fine-tuning, these models fail to explicitly extract helpful knowledge for downstream tasks, leading to unsatisfactory performance. 3) Knowledge graphs are consistently updated (*e.g.,* correcting obsolete knowledge). Existing models cannot adapt to these updates without undergoing retraining. To tackle these challenges, we propose a knowledge retriever that can accurately predict potential knowledge for new proteins, and thus align them with PKGs. This allows the pre-training and fine-tuning stages to directly integrate with knowledge through a unified modeling process, thus unifying the optimization objectives and seamlessly adapting to knowledge updates.

**Generating Candidate Embeddings.** We regard the GO entities in protein knowledge graphs as retrieval candidates. To achieve more accurate and stable retrieval, we integrate the neighboring structure information of each GO entity $g_m$ and generate its candidate embedding as

$$\mathbf{c}_m = \text{MLP}_{aggregation}([\text{MLP}_G(\mathbf{g}_m) : \text{MLP}_G(\mathbf{g}_m^{go}) : \text{MLP}_P(\mathbf{g}_m^{prot})]), \tag{7}$$

where $\mathbf{g}_m$ is the stored embedding of $g_m$. We use $\mathbf{g}_m^{go}$ to incorporate the information of neighboring GO entities of $g_m$, defined as $\mathbf{g}_m^{go} = \frac{1}{|\mathcal{N}_{go}(g_m)|} \sum_{g_k \in \mathcal{N}_{go}(g_m)} \mathbf{g}_k$. Similarly, $\mathbf{g}_m^{prot}$ is used to incorporate the information of $g_m$'s neighboring proteins, defined as $\mathbf{g}_m^{prot} = \frac{1}{|\mathcal{N}_{prot}(g_m)|} \sum_{p_k \in \mathcal{N}_{prot}(g_m)} \mathbf{p}_k$. $\mathcal{N}_{go}(g_m)$ and $\mathcal{N}_{prot}(g_m)$ are respectively the 1-hop neighboring GO entities and 1-hop neighboring proteins of $g_m$. All of $\text{MLP}_{aggregation}$, $\text{MLP}_G$, and $\text{MLP}_P$ are trainable multi-layer perceptrons.

**Retrieval Process.** For each new protein $p_n$, we use a frozen ProtBert model to generate its query embedding as $\mathbf{q}_n = \text{MLP}_P(\text{MEAN}(ProtBert(s_n)))$ where $s_n$ is the amino acid sequence of $p_n$. Intuitively, we can traverse the relation set $R$ and the GO entity set $V_{go}$ to find potential knowledge for $p_n$. However, the time consumption of this strategy is unacceptable because of the large size of $V_{go}$ (*i.e.,* 47K in ProteinKG25). Fortunately, we observed that each relation only connects with several specific GO entities in PKGs, inspiring us to reduce the retrieval complexity by finding relation-GO combinations. Specifically, for relation $r_m \in R$, we construct its candidate GO entity set as $\mathcal{E}(r_m) = \{g_m | (p_x, r_m, g_m) \in F\}$. During retrieval, we traverse each $r_m \in R$ and use each of its corresponding candidate GO entity $g_m \in \mathcal{E}(r_m)$ to construct the candidate knowledge $(p_n, r_m, g_m)$. Then we use the TransE objective Bordes et al. (2013) to score $(p_n, r_m, g_m)$ as

$$\mathbb{S}(p_n, r_m, g_m) = ||\mathbf{q}_n + \tilde{\mathbf{r}}_m - \mathbf{c}_m||_1, \tag{8}$$

where $\tilde{\mathbf{r}}_m = \text{MLP}_{rel}(\mathbf{r}_m)$. Finally, we rank all the candidate knowledge based on their scores, and then add the top-$K$ candidate knowledge into $G$ to align new protein $p_n$ with the knowledge graph.

**Training Strategy.** We use triplets $(p_i, r_i, g_i) \in F$ as valid knowledge and by minimizing a margin-based ranking criterion, we hope that valid knowledge can receive lower scores than invalid knowledge. The training objective is defined as

$$\mathcal{L}_{margin} = \text{MAX}(0, \mathbb{S}(p_i, r_i, g_i) - \mathbb{S}(p_i, r_i, g_j) + \gamma), \tag{9}$$

where $\text{MAX}$ is the maximum operation and $\gamma$ is a hyper-parameter used to control the distance between valid and invalid knowledge. $(p_i, r_i, g_j) \notin F$ is invalid knowledge constructed by perturbing $g_i$ in $(p_i, r_i, g_i)$ with a random GO entity $g_j$. Since the retrieval process needs to match information from different modalities (*i.e.,* text descriptions and amino acid sequences), we further propose a cross-modal matching loss to unify the semantic space of embeddings from different modalities as

$$\mathcal{L}_{match} = \text{MAX}(0, ||\text{MLP}_G(\mathbf{g}_i) - \text{MLP}_P(\mathbf{g}_i^{prot})||_1 - ||\text{MLP}_G(\mathbf{g}_i) - \text{MLP}_P(\mathbf{g}_j^{prot})||_1 + \gamma), \tag{10}$$

where $\mathbf{g}_j^{prot}$ is the neighboring protein embedding of a randomly sampled GO entity $g_j$. This loss forces the text modality information $\text{MLP}_G(\mathbf{g}_i)$ of $g_i$ is closer to its corresponding neighboring protein information $\text{MLP}_P(\mathbf{g}_i^{prot})$ (*i.e.,* amino acid sequence modality) than other protein information $\text{MLP}_P(\mathbf{g}_j^{prot})$. After jointly optimizing $\mathcal{L}_{margin}$ and $\mathcal{L}_{match}$, the knowledge retriever can accurately predict the potential knowledge for new proteins, enabling its effective alignment with PKGs.

### 3.2.2 TASK-ORIENTED FINE-TUNING

After being aligned with PKGs, new proteins can be uniformly encoded with the enhancement of knowledge following Equations (1)-(5), and any related knowledge updates will be perceived when constructing virtual tokens, as they can access the latest version of the PKG to extract relevant knowledge and structures. Then, the downstream task objectives will be used to fine-tune Kara, enabling the protein encoder to extract task-specific useful knowledge from PKGs via virtual tokens. Note that for each new protein $p_n$, we exclude other new proteins from $\mathcal{N}_1(p_n)$ when constructing structure virtual token $\mathbf{v}_n^p$, to avoid noises.

Moreover, the structure-based regularization can also be seamlessly adapted to the fine-tuning stage. This brings two advantages. 1) Downstream tasks usually lack sufficient training data (Rao et al., 2019). The regularization term can introduce biological function similarities among new proteins as an auxiliary optimization objective, thus effectively avoiding over-fitting. 2) By using this regularization as a unified optimization objective of pre-training and fine-tuning, pre-trained knowledge can avoid being catastrophically forgotten and thus effectively transfer to downstream tasks.

**Complexity.** Compared with vanilla protein language models, the additional time complexity of Kara only stems from the virtual tokens and the retrieval process. The two virtual tokens let the encoding complexity become $O((|S| + 2)^2 d)$ from $O(|S|^2 d)$, where $|S|$ is the length of amino acid sequences. Thanks to the proposed strategy of finding relation-GO combinations, the time complexity of retrieving potential knowledge for a new protein is only $O(|R|k_{max})$, where $|R|$ is the size of the relation set in the PKG and $k_{max}$ is the maximum size of the candidate GO entity sets for relations. $k_{max}$ is much smaller than the size of the GO entity set in the protein knowledge graph (*e.g.,* In proteinKG25, $k_{max}$ is about 2K and the size of the GO entity set is 47K).

Table 2: Performance comparisons in the amino acid contact prediction task, where $seq$ indicates the number of amino acids between two selected amino acids. P@L, P@L/2, and P@L/5 denote the precision calculated upon top L (i.e., L most likely contacts), top L/2, and top L/5 predictions, respectively. The best results are **bolded** and the second best results are underlined.

| Models | $6 \leq seq \leq 12$ | | | $12 \leq seq \leq 24$ | | | $24 \leq seq$ | | |
|---|---|---|---|---|---|---|---|---|---|
| | P@L | P@L/2 | P@L/5 | P@L | P@L/2 | P@L/5 | P@L | P@L/2 | P@L/5 |
| LSTM | 0.26 | 0.36 | 0.49 | 0.20 | 0.26 | 0.34 | 0.20 | 0.23 | 0.27 |
| ResNet | 0.25 | 0.34 | 0.46 | 0.28 | 0.25 | 0.35 | 0.10 | 0.13 | 0.17 |
| Transformer | 0.28 | 0.35 | 0.46 | 0.19 | 0.25 | 0.33 | 0.17 | 0.20 | 0.24 |
| ProtBert | 0.30 | 0.40 | 0.52 | 0.27 | 0.35 | 0.47 | 0.20 | 0.26 | 0.34 |
| ESM-1b | 0.38 | 0.48 | 0.62 | 0.33 | 0.43 | 0.56 | 0.26 | 0.34 | 0.45 |
| ESM-2 | 0.40 | 0.50 | 0.62 | 0.35 | 0.44 | 0.56 | 0.27 | 0.35 | 0.45 |
| OntoProtein | 0.37 | 0.46 | 0.57 | 0.32 | 0.40 | 0.50 | 0.24 | 0.31 | 0.39 |
| KeAP | 0.41 | 0.51 | 0.63 | 0.36 | 0.45 | 0.54 | 0.28 | 0.35 | 0.43 |
| Kara | **0.45** | **0.55** | **0.65** | **0.39** | **0.48** | **0.59** | **0.31** | **0.39** | **0.48** |

## 4 EXPERIMENTS AND ANALYSES

We evaluate the generalization ability of Kara in 6 representative downstream tasks, including amino acid contact prediction, homology detection, stability prediction, protein-protein interaction identification, binding affinity prediction, and semantic similarity inference. Ablation studies, hyperparameter studies, and analysis of the knowledge retriever are also provided. The detailed task descriptions are provided in Appendix B. Experimental settings and implementation details are provided in Appendix C. We run each experiment independently three times and report the average results. Codes and datasets are at `https://anonymous.4open.science/r/Kara-1DB8/`.

### 4.1 AMINO ACID CONTACT PREDICTION

**Overview.** This task aims to predict whether two amino acids within a protein are in contact, which is a token-level classification task (Rao et al., 2019). Following Zhou et al. (2023), we use variants of LSTM, ResNet, and Transformer proposed by the TAPE (tasks assessing protein embeddings) benchmark (Rao et al., 2019), pre-trained language models ProtBert (Ahmed et al., 2020), ESM-1b (Rives et al., 2021), and typical knowledge-enhanced model OntoProtein (Zhang et al., 2022) as baselines. The state-of-the-art knowledge-enhanced model KeAP (Zhou et al., 2023) and the recent powerful protein language model ESM-2-30t (Lin et al., 2023) are also used for comparison.

**Results.** As shown in Table 2, Kara outperforms existing models by large margins in all short- ($6 \leq seq \leq 12$), medium- ($12 \leq seq \leq 24$), and long-range ($24 \leq seq$) contact predictions, achieving on average 9.5% and 11.0% improvements in the P@L and P@L/2 metrics. Compared with the state-of-the-art knowledge-enhanced model KeAP, Kara consistently surpasses it, especially in challenging long-range predictions. This is due to Kara's use of contextualized virtual tokens, which allows each amino acid token to explicitly extract task-oriented knowledge information from protein knowledge graphs, thus producing knowledge-contextualized token embeddings with more information. However, KeAP fails to incorporate knowledge during the fine-tuning stage.

### 4.2 PROTEIN-PROTEIN INTERACTION IDENTIFICATION

**Overview.** The protein-protein interaction (PPI) identification task aims to predict the interaction types of protein pairs, which is a sequence-level multi-label classification problem. Our experiments are performed on three widely-used datasets SHS27K (Chen et al., 2019), SHS148K (Chen et al., 2019), and STRING (Lv et al., 2021), where 7 types of interactions are included. Following Zhang et al. (2022), we use DPPI (Hashemifar et al., 2018), DNNPPI (Li et al., 2018), PIPR (Chen et al., 2019), and GNN-PPI (Lv et al., 2021) as four baselines. The LM baselines include ProtBert, ESM-1b, and ESM-2. The knowledge-enhanced model baselines include KeAP and OntoProtein.

**Results.** Experimental results are shown in Table 3. We can see that Kara outperforms baselines on nearly all datasets, highlighting its effectiveness in accurately understanding the relationships between protein sequences. An interesting observation is that the performance gains of KeAP compared with OntoProtein are very small on the STRING dataset. It is suggested in Zhou et al. (2023) that this is because the large number of fine-tuning data in the STRING dataset reduces the impact of

Table 3: Performance comparisons in the protein-protein interaction identification task. BFS (breadth-first search) and DFS (depth-first search) indicate the strategies used to generate test sets on three datasets. We use the F1 score as the evaluation metric.

| | SHS27K | | | SHS148K | | | STRING | | |
|---|---|---|---|---|---|---|---|---|---|
| **Models** | BFS | DFS | Avg | BFS | DFS | Avg | BFS | DFS | Avg |
| DNN-PPI | 48.09 | 54.34 | 51.22 | 57.40 | 58.42 | 57.91 | 53.05 | 64.94 | 59.00 |
| DPPI | 41.43 | 46.12 | 43.77 | 52.12 | 52.03 | 52.08 | 56.68 | 66.82 | 61.75 |
| PIPR | 44.48 | 57.80 | 51.14 | 61.83 | 63.98 | 62.91 | 55.65 | 67.45 | 61.55 |
| GNN-PPI | 63.81 | 74.72 | 69.27 | 71.37 | 82.67 | 77.02 | 78.37 | 91.07 | 84.72 |
| ProtBert | 70.94 | 73.36 | 72.15 | 70.32 | 78.86 | 74.59 | 67.61 | 87.44 | 77.53 |
| ESM-1b | 74.92 | 78.83 | 76.88 | 77.49 | 82.13 | 79.31 | 78.54 | 88.59 | 83.57 |
| ESM-2 | 75.05 | 79.55 | 77.30 | 77.19 | 83.34 | 80.26 | 81.32 | 89.19 | 85.30 |
| OntoProtein | 72.26 | **78.89** | 75.58 | 75.23 | 77.52 | 76.38 | 76.71 | 91.45 | 84.08 |
| KeAP | 78.58 | 77.54 | 78.06 | 77.22 | 84.74 | 80.98 | 81.44 | 89.77 | 85.61 |
| Kara | **81.18** | 78.85 | **80.01** | **79.62** | **86.02** | **82.82** | **82.73** | **92.46** | **87.59** |

knowledge modeling in pre-training. In contrast, Kara incorporates knowledge modeling in both the pre-training and fine-tuning stages, thus avoiding catastrophically forgetting pre-trained knowledge.

## 4.3 HOMOLOGY DETECTION AND STABILITY PREDICTION

**Overview.** Homology detection aims to predict the remote homology of protein, which is a sequence-level classification task. We follow the datasets and experimental settings of Hou et al. (2018), and ask the model to predict the right fold type of protein from 1,195 different types. We report average accuracy on the fold-level held-out set. Stability prediction aims to predict the intrinsic stability of a protein, which is a sequence-level regression task. Following Rocklin et al. (2017), we use Spearman's rank correlation scores to evaluate the model performance. The same baselines are used as in Table 2.

Table 4: Performance comparisons in the protein homology detection and stability prediction tasks.

| **Models** | Homology | Stability |
|---|---|---|
| LSTM | 0.26 | 0.69 |
| ResNet | 0.17 | 0.73 |
| Transformer | 0.21 | 0.73 |
| ProtBert | 0.29 | 0.78 |
| ESM-1b | 0.11 | 0.77 |
| ESM-2 | 0.13 | 0.80 |
| OntoProtein | 0.24 | 0.75 |
| KeAP | 0.29 | 0.80 |
| Kara | **0.32** | **0.83** |

**Results.** As illustrated in Table 4, existing knowledge-enhanced models (*i.e.,* OntoProtein and KeAP) cannot outperform traditional language models in this task. Previous works (Zhang et al., 2022) attributed this failure to the lack of sequence-level objectives during pre-training. Instead, using structure-based regularization, Kara incorporates the knowledge-based relevance (*i.e.,* function similarity) among proteins as a unified sequence-level optimization objective in both pre-training and fine-tuning stages, thus achieving better performance.

## 4.4 PROTEIN-PROTEIN BINDING AFFINITY PREDICTION

**Overview.** This task aims to map each pair of proteins to a real value to indicate their binding affinity changes, which is a sequence-level regression task. Following Unsal et al. (2022), we use Bayesian ridge regression to the element-wise multiplication of protein embeddings for predicting the binding affinity. The SKEMPI dataset (Moal & Fernández-Recio, 2012) is used and the performance is reported based on the mean square error of 10-fold cross-validation. We use the same baselines as recent works (Zhou et al., 2023), additionally with KeAP and ESM-2.

Table 5: Performance comparisons in the protein-protein binding affinity prediction.

| **Models** | Affinity ($\downarrow$) |
|---|---|
| PIPR | 0.63 |
| ProtBert | 0.58 |
| ESM-1b | 0.50 |
| ESM-2 | 0.50 |
| OntoProtein | 0.59 |
| KeAP | 0.52 |
| Kara | **0.50** |

**Results.** As shown in Table 5, all of the existing knowledge-enhanced models fail to outperform ESM-1b. This is because protein structure features play a vital role in this task (Unsal et al., 2022), and the existing models overlook the modeling of protein structures but ESM-1b can achieve it via its network architecture. Kara can achieve competitive performance with ESM-1b because the protein knowledge graph contains the description of the structure properties of proteins, and Kara can directly inject such knowledge information into protein embeddings via the virtual tokens.

Table 7: Ablation study and performance of variants.

| Tasks | Concate ($6 \leq seq \leq 12$) | PPI (STRING) | Homology | Stability | Affinity ($\downarrow$) |
|---|---|---|---|---|---|
| w/o contextualized virtual tokens | 0.42 | 85.16 | 0.28 | 0.81 | 0.55 |
| w/o structure-based regularizations | 0.43 | 86.49 | 0.30 | 0.80 | 0.52 |
| Retrieval based on the protein sequence similarities | 0.43 | 85.33 | 0.29 | 0.79 | 0.57 |
| Kara | **0.45** | **87.59** | **0.32** | **0.83** | **0.50** |

## 4.5 SEMANTIC SIMILARITY INFERENCE

**Overview.** This task evaluates models' ability to extract the biomolecular functional similarity among proteins. Following Unsal et al. (2022), we use biological process (BP) and cellular component (CC) to divide protein attributes into two groups and calculate the Lin similarity in each group as the ground-truth similarity. We then calculate the Manhattan similarity between protein embeddings as the prediction. The Spearman's rank correlation between these similarities is calculated as the metric. We include another powerful protein language model MSA Transformer (Rao et al., 2021) as baseline.

Table 6: Performance in the semantic similarity inference task.

| Models | BP | CC |
|---|---|---|
| MSA Transformer | 0.31 | 0.30 |
| ProtBert | 0.35 | 0.36 |
| ESM-1b | **0.42** | 0.37 |
| ESM-2 | 0.41 | 0.39 |
| OntoProtein | 0.36 | 0.36 |
| KeAP | 0.41 | 0.40 |
| Kara | 0.41 | **0.41** |

**Results.** Table 6 shows that Kara outperforms existing knowledge-enhanced models on both BP and CC. This can be attributed to the explicit incorporation of the information of GO entities in Kara, which describes the functionality of proteins. Kara is unable to outperform ESM-1b on BP may be because of the larger number of parameters of ESM-1b. However, it can still outperform the larger model ESM-1b on CC, indicating its effectiveness in explicitly incorporating GO entity information.

## 4.6 ANALYSIS OF KARA

**Ablation and Variants.** In Table 7 we study the performance contribution of each component in Kara. We can see that all of the virtual tokens, structure-based regularization, and knowledge retriever are essential for achieving good performance. Specifically, removing contextualized virtual tokens makes Kara unable to incorporate knowledge explicitly, and thus significantly degrades its performance in the protein-protein binding affinity prediction task which requires the property understanding of proteins. After removing structure-based regularization, Kara fails to integrate function similarities into sequence-level protein embeddings and thus results in performance degradation in sequence-level tasks, such as homology detection and stability prediction.

To assess the effectiveness of our proposed knowledge retriever, we compare it to a variant that uses a protein similarity-based retriever. In this variant, we use the frozen ProtBert model to calculate embedding similarities between new proteins and those in the PKG, selecting the top-$K$ similar proteins and using their embeddings as virtual tokens. However, this approach does not outperform Kara. The reason is that similarity-based retrievers struggle to accurately predict associated knowledge (*i.e.*, gene descriptions) for proteins, but proteins with similar sequences can have different functions, so this approach may introduce irrelevant protein information as noise during encoding.

**Hyper-parameter Analysis.** During pre-training, we use the ground-truth knowledge graph structure to construct the virtual tokens. However, in the fine-tuning stage, because the new proteins are not included in the protein knowledge graph, we need to use the knowledge retriever to predict its top-$K$ potential knowledge to construct the virtual tokens for fine-tuning and inference, where $K$ is a hyper-parameter used to control the amount of predicted potential knowledge incorporated. Because the predicted potential knowledge can bring additional information but also inevitable noise, in this part we study how $K$ affects the performance

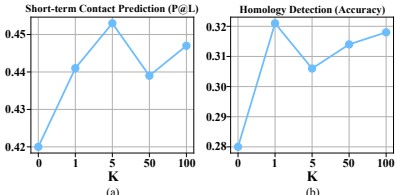

Figure 3: Performance of Kara with different numbers of knowledge $K$.

of Kara. As shown in Figure 3, the performance improves across different tasks when $K$ increases from 0 to 1, showcasing the value of incorporating knowledge into protein representations. As $K$ continues to increase, performance fluctuates due to the introduction of noise from additional knowledge. Nevertheless, it still outperforms the variant without knowledge (*i.e.*, $K$=0), demonstrating Kara's ability to effectively extract useful knowledge for downstream tasks.

### 4.7 ANALYSIS OF KNOWLEDGE RETRIEVER

**Ablation Study.** The accurate retrieval of the knowledge retriever is extremely important for Kara's performance in downstream tasks. Therefore, here we analyze how different components and hyper-parameters affect the retrieval performance of the knowledge retriever. As we mentioned before, the knowledge retriever is trained on

Table 8: Ablation study results of the knowledge retriever.

| Metrics | P@1 | P@5 |
| --- | --- | --- |
| Without structure information | 0.681 | 0.669 |
| Without cross-modal matching | 0.733 | 0.721 |
| Without relation-GO combinations | 0.649 | 0.538 |
| Original | **0.821** | **0.795** |

the ProteinKG25 knowledge graph and we use the randomly sampled 2,000 proteins as the test set to select the best model. During the evaluation, for each test protein $p_t$ we first traverse each relation $r \in R$ to construct query pairs $(p_t, r, ?)$, and then use the knowledge retriever model to score the corresponding candidate knowledge $(p_t, r, g_i^r)$, where $g_i^r$ is the candidate GO entity from $\mathcal{E}(r)$. After traversing all the relations, we rank candidate knowledge based on their scores and calculate the Precision@n (P@n) metric to evaluate the retrieval performance, which indicates how much knowledge on the top-n ranked candidates is valid (*i.e.,* exists in the protein knowledge graph).

**Hyper-parameter Analysis.** In Table 8, without structure information means that we remove the neighbor information in candidate GO embeddings (Equation equation 7), and without cross-modal matching means that the knowledge retriever is only optimized based on $\mathcal{L}_{margin}$. We can see that both of these two components are beneficial to the retrieval performance. Without relation-GO combinations means that for each relation, we use the whole GO entity set as the candidates during retrieval. The worse performance of this variant shows that relation-GO combination strategy can not only reduce the retrieval time consumption, but also help to filter out irrelevant GO candidates and thus improve the retrieval accuracy. As shown in Figure 4, we can see that the higher neighbor sampling number helps to achieve better retrieval performance.

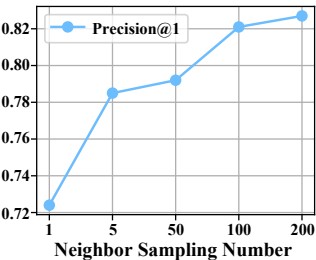

Figure 4: Performance of knowledge retriever with different neighbor sampling numbers.

## 5 RELATED WORK

Protein representation learning has attracted much attention due to the rapid development of pre-trained language models. Existing works treat amino acid sequences as token sequences, and train the language model with either supervision signal (Bepler & Berger, 2019) or self-supervised pre-training objective (Alley et al., 2019; Rao et al., 2019; Xiao et al., 2021; Ahmed et al., 2020; Unsal et al., 2022; Lin et al., 2023; Brandes et al., 2022). However, these approaches ignore factual knowledge (*e.g.,* gene descriptions of proteins), resulting in inferior representations. Recently, OntoProtein (Zhang et al., 2022) is the first attempt to incorporate knowledge graph by proposing a hybrid encoder. KeAP (Zhou et al., 2023) further extends it by performing token-level knowledge exploration via cross-attention module. However, both of them are limited by ignoring knowledge graph structure and task-oriented knowledge modeling. Very recently, GOProteinGNN (Kalifa et al., 2024) explores the benefit of graph structure. However, it still suffers from inconsistent optimization objectives and fails to consider the high-order relationships among proteins. Instead, Kara can explicitly inject high-order knowledge during both the pre-training and fine-tuning stages.

Some models explore incorporating information from other modalities to improve their ability to learn protein representations (Chen et al., 2023a). For example, Otter-Knowledge (Lam et al., 2023) designs knowledge graphs for not only proteins but broadly biomedical concepts. ProtST (Xu et al., 2023) infers protein representations from biomedical texts, but with no graph structure. Our model captures text descriptions together with knowledge graphs for high-order knowledge incorporation.

## 6 CONCLUSION AND FUTURE WORK

We develop a retrieval-augmented language model (Kara) for knowledge-aware protein representation learning, achieving the first unified and direct integration of protein knowledge graphs and protein language models, while considering the high-order relationships within knowledge graphs. Experimental results demonstrate the effectiveness of Kara and its superiority in 6 downstream tasks. A promising future direction is integrating other modalities, such as 3D structures, with knowledge graphs to develop multi-modal, knowledge-aware protein language models.

## REPRODUCIBILITY STATEMENT

Here we detail the efforts that we have made to ensure reproducibility of this work. As shown in Section 4, we provide the anonymous link where the source code of Kara and source data (including both the ProteinKG25 knowledge graph and downstream task datasets) are downloadable. In Appendix B, we provide detailed descriptions of the experimental settings and data processing steps for each downstream task. In Appendix C, we provide detailed descriptions of the experimental environment, backbone selection, hyper-parameter settings, and implementation details (including all of the pre-training and fine-tuning stages, as well as the knowledge retriever). We also provide the official links to pre-trained models and datasets that we used in Kara.

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

# A  DATASET DESCRIPTION

We train the Kara using the ProteinKG25 knowledge graph (Zhang et al., 2022), consistent with previous knowledge-enhanced models to achieve a fair comparison. ProteinKG25 includes about 4.5 million triplets describing relationships between protein and gene ontology (GO) entities, and 100K triplets describing relationships between GO entities. There are 31 kinds of relations, 600K proteins, and 50K GO entities in ProteinKG25. Each GO entity in ProteinKG25 can be a molecule, a cellular component, or a biological process, and each protein in ProteinKG25 has an average of 8.64 relations. Following the strategy provided by Zhou et al. (2023), we removed proteins appearing in the datasets of downstream tasks to avoid data leakage. The raw data of ProteinKG25 can be found in `https://www.zjukg.org/project/ProteinKG25/`.

# B  DOWNSTREAM TASK DEFINITIONS

**Amino Acid Contact Prediction.** This is a pairwise token-level matching task, where each pair of input amino acids $(s^i, s^j)$ from a protein sequence $s$ is mapped to a label $y_{i,j} \in \{0, 1\}$, indicating whether they are in contact or not ($< 8$Å apart). Accurate contact maps can facilitate robust modeling of full 3D protein structure (Kim et al., 2014). Following previous works (Zhou et al., 2023), we use data that comes from ProteinNet (AlQuraishi, 2019) and report precision on the ProteinNet CASP12 test set, which is a standard metric reported in CASP (Moult et al., 2018).

**Protein-protein Interaction Identification.** This is a pairwise sequence-level classification task. Given a pair of proteins $(p_i, p_j)$, the model aims to predict the interaction types $y_{i,j}$ between them. Similar to previous works (Zhou et al., 2023), 7 types of interactions are included in our experiments, which are reaction, binding, post-translational modifications, activation, inhibition, catalysis, and expression. Each protein pair may belong to several types simultaneously so this is a multi-label classification problem. We use three widely-used datasets SHS27K (Chen et al., 2019), SHS148K (Chen et al., 2019), and STRING (Lv et al., 2021) in our experiments, where SHS27K and SHS148K can be regarded as two subsets of STRING, which remove proteins with no more than 50 amino acids or $\geq 40\%$ sequence identity. The F1 score is used as the evaluation metric for this task.

**Homology Detection.** This is a sequence-level classification task where each input protein $p$ is mapped to a label $y \in \{1, 2, ..., 1195\}$ based on its representation generated by protein language models, which indicates its possible protein fold. This task requires the evolutionary understanding of proteins and thus is valuable in microbiology and medicine (*e.g.,* discover new CAS enzymes (Liu et al., 2019)). We follow the previous works and use data from Hou et al. (2018). By holding out entire evolutionary groups from the training set, the model is required to generalize across evolutionary gaps. Same as Hou et al. (2018), we report accuracy on the fold-level heldout set.

**Stability Prediction.** This is a sequence-level regression task. Each input protein $p$ is mapped as a number $y \in \mathbb{R}$, which represents the most extreme conditions under which the protein maintains its structure above a concentration threshold, serving as a proxy for its intrinsic stability. Measuring the stability of proteins is important for finding top candidates from expensive protein engineering experiments (Rao et al., 2019). We use the data provided by Rocklin et al. (2017), where the training set includes proteins from four rounds of experimental design, while the test set contains proteins that are Hamming distance-1 neighbors of the top candidates. We report the Spearman's rank correlation scores on the test set to evaluate the model performance.

**Protein-protein Binding Affinity Prediction.** This is a pairwise sequence-level regression task that maps each pair of proteins $(p_i, p_j)$ as a real value $y \in \mathbb{R}$, indicating the binding affinity changes between them. This task evaluates how well a protein representation can predict changes in binding affinity resulting from protein mutations, thus being valuable for many downstream applications such as drug design (Reidenbach, 2024). Following Unsal et al. (2022), we use Bayesian ridge regression to the element-wise multiplication of protein embeddings for predicting the binding affinity. The SKEMPI dataset (Moal & Fernández-Recio, 2012) is used and the performance is reported based on the mean square error of 10-fold cross-validation.

**Semantic Similarity Inference.** This is a pairwise sequence-level regression task, which evaluates how well protein language models can capture information about biomolecular functional similarity between proteins. In this task, we emphasize the biological process (BP) and cellular component

Table 9: Hyper-parameter settings for different downstream tasks.

| Tasks | Train Steps | Batch Size | $K$ | $\mathcal{L}_{reg}$ | Learning Rate | Gradient Accumulation Step |
|---|---|---|---|---|---|---|
| Contact | 30,000 | 1 | 5 | False | 3e-5 | 8 |
| Homology | 2,200 | 2 | 1 | True | 4e-5 | 16 |
| Stability | 4,800 | 5 | 5 | True | 1e-5 | 16 |

(CC) categories similar to previous works (Unsal et al., 2022). We first use BP and CC to divide protein attributes into two groups and calculate the Lin similarity in each group as the ground-truth similarity. We then calculate the Manhattan similarity between protein embeddings as the prediction. The Spearman's rank correlation between these similarities is calculated as the metric.

# C   EXPERIMENTAL DETAILS

**Experimental Settings.** Same as previous knowledge-enhanced protein language models such as KeAP and OntoProtein, we use the ProtBert model [2] as the backbone of the protein encoder within Kara for a fair comparison. The text descriptions of GO entities and relations are encoded by the PubMedBert model [3], which is also consistent with previous works. While generating the pre-trained embeddings of items in the protein knowledge graph (see Section 2), we represent each item as averaging the embeddings of its amino acid or word tokens. Our model is implemented with Python and we refer to the official code released by Zhou et al. (2023) to implement the downstream task experiments. All tasks use standard datasets and metrics, consistent with previous works, to ensure a fair comparison. Note that since the train/valid/test set splittings of SHS27K, SHS148K, and STRING datasets are not provided, we use the official code released by Lv et al. (2021) to split each dataset with three different random seeds, and the average performance of each dataset is reported. All the experiments are conducted on NVIDIA A40 with 48 GB memory.

**Pre-training Implementation Details.** In the pre-training stage, we set the protein encoder within Kara (*i.e.,* a PortBert model) as full-parameter trainable similar to previous works (Zhang et al., 2022). We only use proteins and knowledge preserved in the ProteinKG25 knowledge graph to pre-train Kara, where the maximum token length is set as 1024 for proteins and 512 for text descriptions. For each protein, we randomly select 10 knowledge and 10 high-order connected proteins respectively from $\mathcal{N}_1$ and $\mathcal{N}_2$ to construct its virtual tokens. The margin $\gamma$ is set as 5 and the number of negative samples is set as 2. We set the batch size to 4 with the maximum number of update steps to 10,000, and the gradient accumulation step to 16. The learning rate is set as 1e-6 and we use AdamW (Loshchilov & Hutter, 2017) for optimization. The weight decay is set as 1e-2.

**Knowledge Retriever Implementation Details.** In the knowledge retriever, we set the sampling number of neighbors during the candidate embedding generation as 100. Similar to the pre-training stage, the maximum token length is 1024 for proteins and 512 for text descriptions. To train the knowledge retriever, we randomly sample 2,000 proteins as well as their associated knowledge from the ProteinKG25 knowledge graph as the test set, and the remaining proteins are used as training data. The best knowledge retriever model is selected based on the Precision@5 metric on the test set. We train the knowledge retriever with the Adam optimizer (Kingma & Ba, 2015). The number of training epochs is set as 500 with the batch size as 100, and we use the early stopping strategy with a patience of 5. The learning rate is set as 1e-3 and the negative sampling number is set as 20. The margin $\gamma$ is also set as 5. Note that we only train the parameters within MLPs and the embeddings of items in the protein knowledge graph are frozen, thus making our knowledge retriever seamlessly generalize to knowledge updates. During inference, we rank all the candidate knowledge for a new protein based on their scores $\mathbb{S}$ (lower is better), and then select top-$K$ knowledge to add to the protein knowledge graph, where $K \in \{1, 5, 50, 100\}$.

**Fine-tuning Implementation Details.** In the fine-tuning stage, we freeze the knowledge projector $MLP_{knowledge}$ and the structure projector $MLP_{structure}$, and only optimize the parameters within the protein encoder for downstream tasks. Note that the protein-protein interaction identification, the protein-protein binding affinity prediction, and the semantic similarity inference tasks do not need fine-tuning and we directly use the pre-trained Kara to encode proteins for these tasks. For the

---

[2]`https://huggingface.co/Rostlab/prot_bert`
[3]`https://huggingface.co/microsoft/BiomedNLP-BiomedBERT-base-uncased-abstract-fulltext`

structure-based regularization term, we still set the margin $\gamma$ as 5 and the number of negative samples as 2. Different downstream tasks require various fine-tuning hyper-parameters and we summarize them in Table 9. Additionally, we follow the implementations in GNN-PPI (Lv et al., 2021) for PPI prediction, where the number of epochs is 600 and batch size is 2048. The learning rate is set as 1e-3 for the SHS27K dataset and 1e-4 for the SHS148K and STRING datasets. We follow the implementations in PROBE (Unsal et al., 2022) for the binding affinity prediction and semantic similarity inference tasks.

