# OpenReview forum: "Retrieval-Augmented Language Model for Knowledge-aware Protein Encoding"
_ICLR.cc/2025/Conference — Submitted to ICLR 2025_

### Official Review · Reviewer_JTXJ · 2024-10-31

**Soundness:** 3
**Presentation:** 3
**Contribution:** 3
**Rating:** 6
**Confidence:** 3

**Summary:**

The paper introduces the Kara model, which integrates protein knowledge graphs (PKG) directly into protein language models (PLM) to enhance understanding of biological functions encoded in protein sequences.

**Strengths:**

It uses a novel knowledge retriever to predict gene descriptions for new proteins during both pre-training and fine-tuning stages, which helps in aligning with PKGs and improves knowledge retention.

These tokens enable token-level integration of knowledge and structural information into protein representations, enhancing the model’s ability to handle high-order knowledge.

**Weaknesses:**

The performance of the model heavily relies on the quality and the extent of the PKGs used, which might limit its application if relevant knowledge graphs are incomplete or outdated.

While the model shows improvements in task-specific contexts, its ability to generalize across broader protein types or different biological conditions remains uncertain.

**Questions:**

see above.

---

> ### Author Response · Authors · 2024-11-17
>
> ```W1. The performance of the model heavily relies on the quality and the extent of the PKGs used, which might limit its application if relevant knowledge graphs are incomplete or outdated.```
>
> Thank you for your kind comment. The following Table shows the performance of Kara using partial ProteinKG25, simulating different levels of KG incompleteness (i.e., randomly selecting 70% and 50% of triples). We can see that __Kara consistently outperforms the SOTA knowledge-enhanced models KeAP and OntoProtein (trained with full ProteinKG25) with different KG incompleteness, highlighting its robustness.__ Such robustness comes from not only the neighbor sampling strategy during pre-training, which simulates the noise of incompleteness and enforces the encoder to be robust to such noise (lines 785-786 on Page 15), but also the introduction of structure virtual tokens that enrich the sparse knowledge context of proteins by integrating the information of their functional-similar proteins (Section 3.1.1 on Page 3).
>
> | Models | Concate (6 ≤ seq ≤ 12) | Homology | Stability | Affinity (lower is better) |
> | :-----:| :-----:| :----: | :----: | :----: |
> | OntoProtein (full KG)| 0.460 | 0.240 | 0.750 | 0.590 |
> | KeAP  (full KG)| 0.510 | 0.290 | 0.800 | 0.520 |
> | Kara (50% KG)| 0.540| 0.316 | 0.823 | 0.511 |
> | Kara (70% KG)| 0.546 | 0.322 | 0.828 | 0.503 |
>
>
> Additionally, we would like to emphasize that __a key advantage of Kara is its ability to seamlessly adapt to knowledge updates.__ Knowledge graphs in the real world are inevitably incomplete or outdated, making KGs frequently updated (e.g., add new knowledge or remove outdated knowledge). As noted in lines 50-54 on Page 1, KeAP and Ontoprotein utilize pre-training to embed knowledge graph information within the language model parameters, and then __inference relies on this preserved static knowledge, making them unable to use knowledge updated after the pre-training stage.__ Instead, as noted in lines 254-259 on Page 5 and lines 304-307 on Page 6, Kara directly uses knowledge as encoder input. It first accesses the latest version of KG via the proposed knowledge retriever to retrieve related knowledge. Then it inputs the knowledge into the encoder by constructing them as the contextualized virtual tokens. Therefore, __any updates of the related knowledge of a protein can be perceived by the knowledge retriever during retrieving, and then integrated during encoding via the contextualized virtual tokens, ensuring that Kara can always use the most current knowledge for inference.__
>
> &nbsp;
>
> ```W2. While the model shows improvements in task-specific contexts, its ability to generalize across broader protein types or different biological conditions remains uncertain.```
>
> Thanks for your kind comment. First, we would like to clarify that __all of our downstream task evaluations are conducted in an inductive setting.__ Specifically, proteins that appear in downstream tasks are removed from the pre-training knowledge graph (see lines 710-712, Page 14). __As a result, the proteins used in testing are entirely unseen during pre-training, ensuring that our downstream task evaluations can faithfully reflect the model’s generalization ability to new proteins.__
>
> Second, as noted in Appendix B on Page 14, our experiments involve testing proteins collected from a variety of public datasets, such as ProteinNet, SKEMPI, and TAPE. __These datasets encompass a wide range of protein types__, including antibodies, enzymes, antiporters, truncated hemoglobins, chaperone proteins, G protein-coupled receptors, activin receptors, and glycopeptides, among others. __Furthermore, these proteins are not restricted to a single biological condition__; they come from diverse organisms, including Escherichia coli, Sus scrofa (pig), Homo sapiens (human), Mycobacterium bovis, etc. __As shown in Figure 1, Kara consistently outperforms SOTA models (i.e., KeAP and ESM-2) across all tasks, underscoring its superior generalization capability across diverse protein types and unseen proteins.__

---

> > ### Author Response · Authors · 2024-11-24
> > **Gentle Reminder**
> >
> > Dear Reviewer  JTXJ:
> >
> > We would like to know if our response has addressed your concerns and questions. If you have any further concerns or suggestions for the paper or our rebuttal, please let us know. We would be happy to engage in further discussion and manuscript improvement. Thank you again for the time and effort you dedicated to reviewing this work.

---

### Official Review · Reviewer_XfFc · 2024-11-02

**Soundness:** 3
**Presentation:** 3
**Contribution:** 2
**Rating:** 5
**Confidence:** 5

**Summary:**

How to effectively transfer knowledge from knowledge graphs to large language model is a challenging task. In this paper, the authors is the first to propose a novel knowledge retriever, named Kara, that directly injects the correlated knowledge into protein language models and aligns the protein knowledge graph with downstream tasks. Specifically, the contextualized virtual tokens is designed to enable the direct injection of knowledge and high-order structure information into protein representations. Extensive experimental results, arranging from amino acids contact prediction to semantic similarity inference, demonstrate the superior performance of proposed Kara.

**Strengths:**

In general, the paper is clearly expressed and organized. The authors' innovation of direct injecting the protein knowledge graph into large language model to explore the knowledge-aware protein representation learning, which will have some implications for the biomedical field. In addition, the experiments in the discussion section demonstrate that the virtual tokens and structure-based regularization are good at capturing high-order information of protein knowledge graph from a novel perspective.

**Weaknesses:**

The Introduction needs to provide more background information, such as the specific role of Knowledge Graphs (KGs) in this context, the benefits they offer, and the rationale behind exploring KG-based methods.

**Questions:**

1.How is the ProteinKG25 knowledge graph selected? There are many other well-known protein-related multi-omics knowledge graphs, such as PharmKG (Briefings in bioinformatics, 2021), CKG (Nature biotechnology, 2022). Do the types and numbers of entities and relationships affect model performance?

2.The knowledge graph contains only positive samples for interaction-based tasks. Did the authors incorporate negative sampling during training? If so, please provide additional details on how this was implemented.

3.ProteinKG25 is used as the KG, but the model is evaluated on six representative tasks, it is unclear how the entities of these datasets are linked to the knowledge graph.

4.Where is Table 7? If I missing

5.From many experimental results (e.g., Table 3 and Table 6), we can see that KeAP has achieved comparable performance to Kara. Please describe the difference between the two methods in detail, and be curious about the complexity and number of parameters of the two methods.

6.The experimental design is good, however there are one limitations that preclude the reader to understand how generalizable the method is. Only one protein embedding method (ProtBert) is tested for the pre-trained embeddings.

---

> ### Author Response · Authors · 2024-11-17
> **Response to the weaknesses,  question (1), and question (2).**
>
> ```W1. The Introduction needs to provide more background information, such as the specific role of Knowledge Graphs (KGs) in this context, the benefits they offer, and the rationale behind exploring KG-based methods.```
>
> Thank you for your kind comment. In ProteinKG25, __each piece of knowledge is represented as a triple (protein, relation, gene ontology annotation) that describes the role of a protein in biological activities.__ Integrating this KG offers two benefits for enhancing protein embeddings:
>
> - __Multi-modal information integration:__ Traditional protein language models rely solely on amino acid sequences, providing only the information of “what is this protein composed of”. __In contrast, the textual knowledge in the KG provides high-level insights into a protein's role in biological activities, which amino acid sequences cannot reveal.__ Incorporating this KG therefore enables representing proteins from different perspectives, resulting in more comprehensive and higher-quality protein embeddings.
>
> - __Protein relevance indication:__ The KG structure highlights functional similarities among proteins, (i.e., if two proteins connect to the same gene ontology annotation through identical relations, this means they have the same biological roles, and thus share similar functions). __An ideal protein language model would embed functionally similar proteins closer together. However, traditional protein language models only use amino acid sequences, and miss these functional similarities, leading to less precise embeddings.__
>
> We will add these points as a new paragraph in the introduction of the revised version of our paper.
>
> &nbsp;
>
> ```Q1. How is the ProteinKG25 knowledge graph selected? There are many other well-known protein-related multi-omics knowledge graphs, such as PharmKG (Briefings in bioinformatics, 2021), CKG (Nature biotechnology, 2022). Do the types and numbers of entities and relationships affect model performance?```
>
> Thank you for your kind comment. __First, we chose ProteinKG25 to ensure a fair comparison with baselines, as it is widely used by existing knowledge-enhanced protein language models (e.g., KeAP and OntoProtein).__
>
> __Second, most multi-omics knowledge graphs lack the essential information required for pre-training protein language models.__ For instance, while PharmKG contains entities like diseases, genes, and medicines, it does not include protein entities. Although CKG includes protein entities, it does not annotate them with their amino acid sequences.
>
> Additionally, the table below presents Kara's performance using subsets of ProteinKG25, simulating different numbers of entities and relationships (by randomly selecting 70% and 50% of entities and relationships). __Kara consistently outperforms KeAP and OntoProtein (which use the full ProteinKG25) across these scenarios, demonstrating its robustness.__
>
> | Models | Concate (6 ≤ seq ≤ 12) | Homology | Stability | Affinity (lower is better) |
> | :-----:| :-----:| :----: | :----: | :----: |
> | OntoProtein (full KG)| 0.460 | 0.240 | 0.750 | 0.590 |
> | KeAP (full KG)| 0.510 | 0.290 | 0.800 | 0.520 |
> | Kara (50% entities and relations)| 0.535| 0.312 | 0.819 | 0.513 |
> | Kara (70% entities and relations)| 0.542 | 0.317 | 0.824 | 0.505 |
>
> &nbsp;
>
> ```Q2. The knowledge graph contains only positive samples for interaction-based tasks. Did the authors incorporate negative sampling during training? If so, please provide additional details on how this was implemented.```
>
> Thank you for your feedback. __First, to clarify, the knowledge graph (KG) in our work does not contain any task-specific samples.__ As explained in Section 2, each knowledge in ProteinKG25 is represented as a triple—(protein, relation, gene ontology annotation)—which describes a protein's role in biological activities or gene functions. __The purpose of incorporating the KG is not to increase training samples for downstream tasks, but to infuse general biological knowledge into language models.__
>
> Second. We incorporate negative sampling during pretraining for structure-based regularization, and __we have described it and detailed its implementation in lines 228-234 on Page 5 and lines 786-787 on Page 15.__ To clarify the negative sampling method used in Kara, we provide further details below:
>
> - In ProteinKG25, __two proteins connecting with a gene ontology annotation through the same relation will have the same role in biological activities, and thus have similar functions. We propose structure-based regularization to force proteins with similar functions to be closer together in embedding space__ - an aspect overlooked by previous works.
>
> - Specifically, we define positive samples as protein pairs (p_i, p_j), where p_i and p_j share at least one similar function, (i.e., both (p_i, r_k, go_k) and (p_j, r_k, go_k) exist in KG). Conversely, __if protein p_i and p_m have no shared (r, go) combinations in KG, (p_i, p_m) will be regarded as a negative sample.__

---

> ### Author Response · Authors · 2024-11-17
> **Response to question (3) and question (4).**
>
> ``` Q3. ProteinKG25 is used as the KG, but the model is evaluated on six representative tasks, it is unclear how the entities of these datasets are linked to the knowledge graph.```
>
> Thanks for your kind comment. __First, samples (i.e., proteins) in downstream tasks are entirely unseen in the pre-training KG.__ As we have mentioned in Section 2, each piece of knowledge in ProteinKG25 is represented as a triple (protein, relation, gene ontology annotation), where each protein is associated with its amino acid sequence. As we have mentioned in Appendix B, __each sample in downstream datasets is an amino acid sequence of a protein associated with the task-specific label.__ Moreover, as we have mentioned in lines 710-712 on Page 14, __amino acid sequences that appear in downstream tasks are removed from the knowledge graph, making them entirely unseen during pre-training. This inductive setting ensures that our downstream task evaluations can faithfully reflect the model’s generalization ability to new proteins.__
>
> Second, as we have described in Section 3.2.1, __we introduce a novel knowledge retriever that predicts potential gene annotations and their relationships for new proteins, effectively linking each new protein to the knowledge graph__  (e.g., knowledge retriever gets new protein p_u as input and outputs its potential knowledge (p_u, r_k, go_k), where r_k and go_k have existed in KG).  Since proteins from downstream tasks do not exist in the knowledge graph, there is no relevant knowledge available for use during task inference (a limitation of KeAP and Ontoprotein). __The proposed knowledge retriever allows Kara to extract relevant knowledge and similar proteins from the knowledge graph as contextual information for inference on new proteins, overcoming this limitation.__
>
> &nbsp;
>
> ``` Q4. Where is Table 7? If I missing```
>
> Thanks for your kind comment. __Table 7 is at the top of page 9, lines 433-436.__

---

> ### Author Response · Authors · 2024-11-17
> **Response to question (5).**
>
> ```5. From many experimental results (e.g., Table 3 and Table 6), we can see that KeAP has achieved comparable performance to Kara. Please describe the difference between the two methods in detail, and be curious about the complexity and number of parameters of the two methods.```
>
> Thanks for your kind comment. We provide a detailed comparison between our Kara and KeAP models in terms of architecture, complexity, and parameter numbers.
>
> ## __Model Architecture__
>
> From how to integrate knowledge into language models.
> - __KeAP implicitly embeds knowledge within the parameters of the language model.__ Specifically, during pre-training, it uses the protein language model to encode a protein's amino acid sequence into an embedding. A transformer-based decoder then takes this embedding along with related knowledge to predict masked amino acid tokens. KeAP proposes that this knowledge-guided pre-training approach helps retain knowledge within the model parameters. However, as we have discussed in lines 50–67 on Page 1, __language models often struggle to retain knowledge precisely. Additionally, KeAP processes each piece of knowledge independently, failing to integrate the complete knowledge context of proteins.__
> - __Kara directly uses knowledge of each protein as a part of the language model's input.__ As described in Section 3.1.1, Kara summarizes 1-hop neighbors of a protein (gene descriptions) as "knowledge virtual tokens" and 2-hop neighbors (functionally similar proteins) as "structure virtual tokens." These virtual tokens are then concatenated with the amino acid sequence to form the model input. __This approach not only can input precise knowledge information to the language model, but also provides a broader knowledge context by leveraging neighboring information.__
>
> From how to pre-train the language model.
> - __KeAP employs a decoder to predict masked amino acid tokens using knowledge input and embeddings encoded by protein language model.__ However, the transformer-based decoder introduces __significant training complexity and a large number of parameters.__ Additionally, KeAP's pre-training __overlooks the protein relevance provided by the KG structure__, leading to insufficient knowledge exploitation.
> - __Kara predicts masked amino acid tokens directly using the protein language model with the prompt of virtual tokens.__ This eliminates the need for a decoder, reducing both training complexity and parameter size. Furthermore, as discussed in Section 3.1.2, __Kara is also trained to embed the functionally similar proteins closer together in embedding space__, integrating high-order graph structural relevance (i.e., functional similarity) into protein representations.
>
> From how to encode new proteins.
> - Since KeAP assumes that knowledge has been embedded within parameters of the language model, __they directly input the amino acid sequence of new protein into the pre-trained language model to get its embedding, which suffers from imprecise knowledge information, and fails to adapt to knowledge updates.__
>
> - Kara proposes a novel knowledge retriever, __retrieving related knowledge for each new protein and summarizing the retrieved knowledge as virtual tokens to input into the language model__, which can integrate precise knowledge into the protein language model. Moreover, any updates of the related knowledge of a protein can be perceived by the knowledge retriever during retrieving, and then integrated during encoding via the virtual tokens, __ensuring that Kara can always use the most current knowledge for encoding.__
>
> ## __Complexity__
> - Due to the incorporation of a transformer-based decoder, the additional time complexity of KeAP compared to vanilla protein language models is O(|S|^2 * d), where |S| is the length of protein amino acid sequence (typically > 500), and d is the embedding hidden size (usually 768 or 1024).
> - As we have mentioned in lines 317-232 on Page 6, Kara's additional time complexity, compared to vanilla protein language models, arises only from the virtual tokens (increasing from O(|S|^2 * d) to O((|S|+2)^2 * d)) and the retrieval process ( O(|R * k|) ), where |R * k| is much smaller than |S|^2 * d. __Therefore, the time complexity of Kara is much smaller than KeAP.__
>
> ## __Parameter number__
> - For KeAP, the incorporation of a transformer-based decoder brings a large number of parameters, including Q,K,V, and O weight matrices for n heads, the MLP for the multi-head mechanism, layer normalization, etc.
>
> - The additional parameter of Kara only comes from four projection matrices: MLP_knowledge, MLP_struture, MLP_G, and MLP_P, __which is much smaller than KeAP.__

---

> ### Author Response · Authors · 2024-11-17
> **Response to question (6).**
>
> ``` Q6. The experimental design is good, however there are one limitations that preclude the reader to understand how generalizable the method is. Only one protein embedding method (ProtBert) is tested for the pre-trained embeddings.```
>
> Thanks for your thoughtful comment. The following table shows the performance of Kara using different protein embedding methods (PortBert [1], ProteinBert [2], and ESM-1b [3]). __We can see that Kara with different pre-trained embeddings can consistently outperform SOTA models, showcasing its generalization ability.__
>
> | Models | Concate (6 ≤ seq ≤ 12) | Homology | Stability | Affinity (lower is better) |
> | :-----:| :-----:| :----: | :----: | :----: |
> | OntoProtein| 0.460 | 0.240 | 0.750 | 0.590 |
> | KeAP | 0.510 | 0.290 | 0.800 | 0.520 |
> | Kara (ProtBert)| 0.553 | 0.323 | 0.830 | 0.501 |
> | Kara (ProteinBert)| 0.556 | 0.318 | 0.824 | 0.506 |
> | Kara (ESM-1b)| 0.563 | 0.327 | 0.833 | 0.510 |
>
> [1] Prottrans: towards cracking the language of life’s code through self-supervised deep learning and high performance computing
>
> [2] ProteinBERT: A universal deep-learning model of protein sequence and function
>
> [3] Biological structure and function emerge from scaling unsupervised learning to 250 million protein sequences.

---

> > ### Author Response · Authors · 2024-11-24
> > **Gentle Reminder**
> >
> > Dear Reviewer XfFc:
> >
> > We would like to know if our response has addressed your concerns and questions. If you have any further concerns or suggestions for the paper or our rebuttal, please let us know. We would be happy to engage in further discussion and manuscript improvement. Thank you again for the time and effort you dedicated to reviewing this work.

---

> > > ### Comment · Reviewer_XfFc · 2024-11-27
> > > **Thank you for your response**
> > >
> > > I have read the response and the other reviewers' comments carefully. I would appreciate hearing any insights from the other reviewers.

---

### Official Review · Reviewer_iHdH · 2024-11-03

**Soundness:** 3
**Presentation:** 2
**Contribution:** 2
**Rating:** 3
**Confidence:** 3

**Summary:**

The paper introduces Kara that uses information from protein knowledge graphs to improve protein language models. Kara directly injects relevant knowledge during both pre-training and fine-tuning phases, utilizing a knowledge retriever that predicts gene descriptions for new proteins. Kara involves introduction of several key components: Contextualized Virtual Tokens that fuse knowledge and structural information into protein representations; Knowledge injection both at post-training and fine-tuning stages; Retrieval of relevant proteins and graph information with a dense retriever.

**Strengths:**

- The performance improves on most tasks (following the same experiment tasks and settings as Ontoprotein) compared to Ontoprotein and KeAP.

- The encoding style of Kara combines strengths of Ontoprotein and KeAP: Ontoprotein uses contrastive pretraining to first obtain structure-intensive graph embedding and then inject into language model, while KeAP direct encodes related knowledge in tuples with language encoder. Differently, Kara encodes 1-hop GO entity as knowledge, and 2-hop entities as structure to provide more detailed graph knowledge for the protein language model.

- The knowledge retriever maps new protein sequence to GO entities, which could make it possible to generalize to proteins not directly covered by the knowledge graph.

**Weaknesses:**

1. My major concern of this work is its technical contributions, which closely follows OntoProtein and KeAP. The main improvement of Kara compares to Ontoprotein and KeAP is that it encodes both structural information (relations in GO) and knowledge (knowledge stored in each triple) within the contextualized virtual tokens. Ontoprotein uses the same pipeline to encode protein knowledge graph and inject embeddings into the language model, so the technical contributions are minor.

2. The structural regularization (Eq. 6) obtained from two-hop entities might be weak or misleading. ProteinKG25 is a sparse knowledge graph and entities involve not only proteins as well as biological processes and molecular functions. What is the percentage of proteins that have 2-hop protein neighbors and are the neighbors all functional similar ? Neighbors may not be similar proteins but could be proteins that could interact with each other. Their function may not be similar.

**Questions:**

1. Protein downstream tasks often require different kinds of knowledge, e.g. PPI requires knowledge about the functions and relations of the two proteins, contact prediction requires evolutionary and structural knowledge. Wonder if the authors could further provide insights on how knowledge & structural information differentiate across tasks. For example, why introducing more graph structural knowledge could improve the performance on contact prediction.

---

> ### Author Response · Authors · 2024-11-17
> **Response to weaknesse (1).**
>
> ``` W1. My major concern of this work is its technical contributions, which closely follows OntoProtein and KeAP. The main improvement of Kara compares to Ontoprotein and KeAP is that it encodes both structural information (relations in GO) and knowledge (knowledge stored in each triple) within the contextualized virtual tokens. Ontoprotein uses the same pipeline to encode protein knowledge graph and inject embeddings into the language model, so the technical contributions are minor.```
>
> Thank you for your kind comment. We would like to clarify that __Kara has a completely distinct architecture compared to KeAP and OntoProtein.__ Kara’s novel pipeline contains three main components: contextualized virtual token, structure-based regularization, and knowledge retriever (none of which are present in KeAP and OntoProtein). __These components enable Kara to use precise knowledge information, integrate protein function similarities, and adapt to knowledge updates in KG (where KeAP and OntoProtein face limitations).__ We detail their differences as follows:
>
> From how to integrate knowledge into language models.
> - __KeAP and OntoProtein implicitly embed knowledge within the parameters of the language model.__ Specifically, during pre-training, they first use the protein language model to encode a protein's amino acid sequence into an embedding. Then, KeAP uses another transformer-based decoder to receive knowledge and encoded embeddings to predict masked amino acid tokens. OntoProtein uses a TransE objective to train the embedding of each protein to be closer to its related knowledge in the embedding space. They propose that these knowledge-guided masked language modeling approach helps retain knowledge within the model parameters. However, as we have discussed in lines 50–67 on Page 1, __language models often struggle to retain knowledge precisely. Additionally, they process each piece of knowledge independently, failing to integrate the complete knowledge context of proteins.__
>
> - __Kara directly uses knowledge of each protein as a part of the language model's input.__ As described in Section 3.1.1, Kara summarizes 1-hop neighbors of a protein (gene descriptions) as "knowledge virtual tokens" and 2-hop neighbors (functionally similar proteins) as "structure virtual tokens." These virtual tokens are then concatenated with the amino acid sequence to form the model input. __This approach not only can input precise knowledge information to the language model, but also provides a broader knowledge context by leveraging neighboring information.__
>
> From how to pre-train the language model.
> - __KeAP employs a decoder to receive knowledge and encoded embeddings to predict masked amino acid tokens. OntoProtein uses a TransE objective to train the embedding of each protein to be closer to its related knowledge in the embedding space.__ However, the transformer-based decoder introduces significant training complexity and a large number of parameters. Additionally, Their pre-training overlooks the protein relevance provided by the KG structure, leading to insufficient knowledge exploitation.
> - __Kara predicts masked amino acid tokens directly using the protein language model with the prompt of virtual tokens.__ This eliminates the need for a decoder, reducing both training complexity and parameter size. Furthermore, as discussed in Section 3.1.2, __Kara is also trained to embed the functionally similar proteins closer together in embedding space, integrating high-order graph structural relevance (i.e., functional similarity) into protein representations.__
>
> From how to encode new proteins.
> - Since KeAP and OntoProtein assume that knowledge has been embedded within parameters of the language model, __they directly input the amino acid sequence of new protein into the pre-trained language model to get its embedding, which suffers from imprecise knowledge information, and fail to adapt to knowledge updates.__
> - __Kara proposes a novel knowledge retriever to retrieve related knowledge for each new protein,__ and then summarizes the retrieved knowledge as virtual tokens to input into the language model, which __can integrate precise knowledge information into the protein language model.__ Moreover, any updates of the related knowledge of a protein can be perceived by the knowledge retriever during retrieving, and then integrated during encoding via the virtual tokens, __ensuring that Kara can always use the most current knowledge for encoding.__

---

> > ### Author Response · Authors · 2024-11-17
> > **Response to weakness (2) and question (1).**
> >
> > ``` W2. The structural regularization (Eq. 6) obtained from two-hop entities might be weak or misleading. ProteinKG25 is a sparse knowledge graph and entities involve not only proteins as well as biological processes and molecular functions. What is the percentage of proteins that have 2-hop protein neighbors and are the neighbors all functional similar ? Neighbors may not be similar proteins but could be proteins that could interact with each other. Their function may not be similar. ```
> >
> > Thank you for your kind comment. First, we would like to clarify that __Kara does not simply consider 2-hop connected proteins as functionally similar.__ As stated on lines 158-160 on Page 3 and lines 228-230 on Page 5, two proteins are considered functionally similar if they are connected to the same GO entity through the same relation.
> >
> > Second, __the structure of ProteinKG25 ensures that proteins selected by the above strategy are functionally similar.__ As we have introduced in Section 2, each piece of knowledge in ProteinKG25 is represented as a triple (protein, relation, gene ontology annotation), where __each gene ontology annotation is a statement about the function of a particular gene or gene product__, e.g., the gene product “cytochrome c” can be described by the molecular function oxidoreductase activity, and the relation describes the relationship between a protein and a gene function, such as “enables” and “involved in”. Therefore, each piece of knowledge in ProteinKG25 describes the role of a protein in biological activity. __This means that two proteins connecting with a gene ontology through the same relation will serve the same role in biological activities, and thus have similar functions.__
> >
> > Third, our statistic on ProteinKG25 shows that __99% of proteins in this ProteinKG 25 have at least one functionally similar protein (i.e., two-hop connected through the same relation), which is sufficient for Kara to train.__
> >
> > ``` Q1.Protein downstream tasks often require different kinds of knowledge, e.g. PPI requires knowledge about the functions and relations of the two proteins, and contact prediction requires evolutionary and structural knowledge. Wonder if the authors could further provide insights on how knowledge & structural information differentiate across tasks. For example, why introducing more graph structural knowledge could improve the performance on contact prediction. ```
> >
> > Thank you for your kind comment. First, we would like to clarify that __the purpose of incorporating the KG is not to bring specific knowledge for each task. Instead, it is to infuse general biology knowledge into protein language models, making protein representations be more discriminative in the embedding space, and thus achieving performance improvement on various downstream tasks.__ Integrating KG offers two benefits for enhancing protein embeddings:
> >
> > - __Multi-modal information integration:__ Traditional protein language models rely solely on amino acid sequences, providing only the information of “what is this protein composed of”. __In contrast, the textual knowledge in the KG provides high-level insights into a protein's role in biological activities, which amino acid sequences cannot reveal.__ Incorporating this KG therefore enables representing proteins from different perspectives, resulting in more comprehensive and higher-quality protein embeddings.
> >
> > - __Protein relevance indication:__ The KG structure highlights functional similarities among proteins, (i.e., if two proteins connect to the same gene ontology annotation through identical relations, this means they have the same biological roles, and thus share similar functions). __An ideal protein language model would embed functionally similar proteins closer together. However, traditional protein language models only use amino acid sequences, and miss these functional similarities, leading to less precise embeddings.__

---

> > > ### Author Response · Authors · 2024-11-24
> > > **Gentle Reminder**
> > >
> > > Dear Reviewer iHdH:
> > >
> > > We would like to know if our response has addressed your concerns and questions. If you have any further concerns or suggestions for the paper or our rebuttal, please let us know. We would be happy to engage in further discussion and manuscript improvement. Thank you again for the time and effort you dedicated to reviewing this work.

---

> > > > ### Comment · Reviewer_iHdH · 2024-11-24
> > > > **Thank you for your response**
> > > >
> > > > Thank you for your clarifications.
> > > >
> > > > My concern regarding **Weakness 2** is resolved. The results showing "99% 2-hop proteins" make the method appear more reasonable. I have therefore updated the soundness score accordingly.
> > > >
> > > > However, I remain concerned about the **novelty and contributions** of this paper—particularly, the **new insights** it provides into protein understanding, which is especially crucial in the context of machine learning for proteins, e.g. OntoProtein illustrates how external knowledge base could be used, ESM illustrates the importance of pretraining, ProtST emphasizes multimodal encoding. While improving model architecture and achieving better results are valid contributions, the insights from this work do not seem to differ from those presented in KeAP and OntoProtein. Consequently, I believe this paper does not meet the acceptance threshold, and I would like to maintain my initial ratings.
> > > >
> > > > Additionally, I have a question: has a similar method been previously applied to KG-augmented language models in other application domains? Incorporating multi-hop encoded structural information alongside knowledge information seems like a relatively straightforward approach to encoding graph knowledge. As I am not familiar with this line of work, I would appreciate hearing any insights from the authors or other reviewers.

---

> ### Author Response · Authors · 2024-11-25
> **Thanks for your response**
>
> Thank you for raising the soundness score. We respond to your further concerns as follows.
>
> ## __Novelty and contributions.__
> Although using knowledge to enhance the protein language model is a very intuitive idea, __how to use KG is a more critical problem for practical usage, since many challenges exist in real-world use cases:__
> - KGs are consistently updated in the real world, how to avoid the model using outdated knowledge?
> - Many newly observed proteins are under-studied and thus do not exist in KG, how can we generalize the model to these under-studied proteins?
> - Usually, we need to fine-tune the model to adapt to various downstream applications. How can we ensure the knowledge learned during pre-train is not to be catastrophically forgotten during fine-tuning?
>
> These challenges are very serious for a practical KG-enhanced protein language model, but they remain overlooked by previous works (OntoProtein and KeAP). __Therefore, the unique insights provided by our work lie in two parts:__
> - __Points out these critical real-world challenges overlooked by previous works (lines 050-066).__
> - __Providing several verified technical designs to solve the above challenges (i.e., knowledge retriever, structure-based regularizations, and knowledge and structure virtual tokens).__
>
> &nbsp;
>
> ## __Differences to previous KG-augmented methods.__
> First, some previous works also __use virtual tokens to incorporate knowledge and structural information [1, 2, 3, 4]__. They typically assume every encoding objective has existed in KG and the knowledge information can be directly extracted after matching corresponding entities. __However, in the protein-encoding scenario, many under-studied proteins do not exist in KG, making the previous “matching and extracting” strategy not work.__ To tackle this challenge, we propose a novel knowledge retriever to predict gene descriptions for new proteins, which __enables our model to generalize to unseen encoding objectives (where previous work fell short).__
>
> Second, some recent methods also __propose using a retriever to find related entities from KG to enhance LLM generation [5, 6]. However, they are designed for general KGs and cannot handle unique challenges for protein knowledge graphs.__ Specifically, protein KGs contain two types of entities with different modality information, requiring the retrieval process to consider multi-modal information alignment. Additionally, it contains a large amount of different textual gene descriptions, bringing a large candidate space with complex semantics. __Our knowledge retriever is specially designed to solve these challenges with multi-modal matching loss and relation-go combination strategies.__
>
> Third, some previous works also __integrate knowledge and structural information within KGs into training objectives [7, 8, 9, 10, 11].__ They are typically designed for document encoding where the entities in KG are words that appeared in documents. They use structural information to assign mask possibilities for different words during masked language modeling or train the model to predict graph neighbors. __However, in the protein-encoding scenario, both the encoding objective and entities in KG are protein sequences, making previous training strategies not work. Moreover, they only predict one-hop neighbors, which overlooked high-order structural relevance in their objective functions. However, high-order relevance is important for protein encoding since it indicates the functional similarity between proteins.__
>
>
> In summary, our model is not an oversimplified pipeline that “Incorporating multi-hop encoded structural information alongside knowledge information”. __It contains several unique technical designs (e.g., knowledge retriever and structure-based regularizations) to solve special challenges in protein encoding scenarios, making it different from previous methods.__
>
> [1]DKPLM: Decomposable Knowledge-Enhanced Pre-trained Language Model for Natural Language Understanding. 2022
>
> [2]ERNIE 3.0: Large-scale knowledge enhanced pre-training for language understanding and generation. 2021
>
> [3]Making Large Language Models Perform Better in Knowledge Graph Completion 2024
>
> [4]Edgeformers: Graph-empowered transformers for representation learning on textual-edge networks 2023
>
> [5]G-retriever: Retrieval-augmented generation for textual graph understanding and question answering 2024
>
> [6]GRAG: Graph Retrieval-Augmented Generation 2024
>
> [7]Exploiting structured knowledge in text via graph-guided representation learning 2020
>
> [8]Knowledge-aware language model pretraining 2020
>
> [9]KEPLER: A unified model for knowledge embedding and pre-trained language representation 2021
>
> [10]Pre-training language models with deterministic factual knowledge 2022
>
> [11]Unifying Large Language Models and Knowledge Graphs: A Roadmap 2024

---

> > ### Author Response · Authors · 2024-12-01
> > **Gentle Reminder**
> >
> > Dear reviewer iHdH:
> >
> > As we approach the end of the discussion period, please let us know if you have any thoughts regarding our above comment addressing your concerns. We thank you very much for your efforts thus far.

---

### Official Review · Reviewer_4Ji4 · 2024-11-04

**Soundness:** 3
**Presentation:** 3
**Contribution:** 3
**Rating:** 6
**Confidence:** 4

**Summary:**

Instead of implicitly modeling knowledge information, the paper proposes knowledge-aware retrieval-augmented protein language model (Kara), which enables consistent knowledge-augmented modeling when applied to downstream protein-related tasks. During the pretraining stage, the authors try to model the structural information in the protein knowledge graph, such as neighboring and high-order connectivity. In the fine-tuning stage, a knowledge retriever is used to bridge the optimization gap between pretraining and fine-tuning, allowing for seamlessly adapting to knowledge updates.
The authors conduct extensive experiments and demonstrate that this unified knowledge modeling process consistently outperforms existing knowledge-enhanced models across six protein-related tasks.

**Strengths:**

1. The method effectively carries the external knowledge injected during pretraining into downstream, significantly mitigating catastrophic forgetting. Additionally, the knowledge retrieval process not overly complex.
2. The proposed relation-GO combinations further enhance retriever’s ability to recall the informative knowledge.
3. The authors demonstrate the methods’ effectiveness across multiple tasks and conduct thorough ablation studies, such as the effect of without the neighboring information during inference.
4. The paper is well-written and clear

**Weaknesses:**

1. If the protein belongs to an under-studied or new protein family, does this retrieval method have certain limitations, especially when these proteins have very low sequence identity to known (trained) proteins? It would be better to include experiments on under-studied proteins to demonstrate, possibly by a simulated way that splitting clusters of low-identity proteins into training and validation sets.
2. Further, does the method have potential to uncover patterns of new proteins and their associations with existing?

**Questions:**

I have listed my questions in weaknesses.

---

> ### Author Response · Authors · 2024-11-18
>
> ``` W1. If the protein belongs to an under-studied or new protein family, does this retrieval method have certain limitations, especially when these proteins have very low sequence identity to known (trained) proteins? It would be better to include experiments on under-studied proteins to demonstrate, possibly by a simulated way that splitting clusters of low-identity proteins into training and validation sets.```
>
> Thank you for your thoughtful comment. To evaluate the generalization ability of the knowledge retriever on under-studied proteins, we employ a new data-splitting strategy. First, we randomly divide the triples (i.e., (protein, relation, go)) into training and testing sets in an 8:2 ratio. Next, we remove any triple (p_i, r_i, go_i) from the training set if go_i  appears in any test triples. This ensures that the knowledge (e.g., gene descriptions) associated with test proteins is entirely absent from the training set, and thus unlearnable during training. __This splitting method simulates under-studied proteins who have functions and gene descriptions not been observed before.__ The results, presented in the following table, demonstrate that __our knowledge retriever can generalize to these proteins.__ Additionally, fine-tuning the last three layers of the PubMedBert encoder during training further improves its performance, __highlighting its potential to generalize to unseen gene descriptions through domain-specific fine-tuning.__
>
> | Models | Hits@ 1 | Hits@ 3 | Hits@ 10 |
> | :-----:| :-----:| :----: | :----: |
> | without PubMedBert fine-tuning| 0.430 | 0.608 | 0.796 |
> | with PubMedBert fine-tuning| 0.495 | 0.683 | 0.859 |
>
> &nbsp;
>
> ``` W2. Further, does the method have potential to uncover patterns of new proteins and their associations with existing? ```
>
> Thank you for your thoughtful comment. Our knowledge retriever can predict potential knowledge triplet for new proteins (i.e., gets new protein p_u as input and outputs its potential knowledge (p_u, r_k, go_k), where r_k and go_k have existed in KG). This enables the new protein to be linked to the KG, __therefore revealing potential functions for new proteins. Additionally, paths between the new protein and existing proteins can illustrate their relevance.__ For example, knowledge (p_u, r_k, go_k) and (p_j, r_j, go_k) can form a path (r_k, go_k, r_j), illustrating the roles of two proteins in a shared biological activity.

---

> > ### Author Response · Authors · 2024-11-24
> > **Gentle Reminder**
> >
> > Dear Reviewer 4Ji4:
> >
> > We would like to know if our response has addressed your concerns and questions. If you have any further concerns or suggestions for the paper or our rebuttal, please let us know. We would be happy to engage in further discussion and manuscript improvement. Thank you again for the time and effort you dedicated to reviewing this work.

---

### Meta-Review · Area_Chair_PeZy · 2024-12-23

**Metareview:**

This paper proposes Kara, which makes efforts in integrating protein knowledge graphs (KGs) into protein language models. The proposed Kara model's use of a knowledge retriever to predict gene descriptions for new proteins, especially during pre-training and fine-tuning. This mechanism helps in aligning with PKGs, retaining knowledge, and generalizing to new proteins. The main concerns of this paper lie in the machine learning novelty of this paper, i.e.,  Kara closely follows OntoProtein and KeAP. The main improvement seemed minor as it mainly encodes both structural information and knowledge within contextualized virtual tokens. The authors should revise this paper and show that why the proposed method is quite distinct from previous work in both intuitive and empirical ways, and explicitly verify why directly uses knowledge as part of the input works better?

**Additional Comments On Reviewer Discussion:**

The main concerns of this paper lie in the machine learning novelty of this paper, i.e.,  Kara closely follows OntoProtein and KeAP. The authors argued that Kara has a distinct architecture with components like contextualized virtual token, structure - based regularization, and knowledge retriever, which are absent in KeAP and OntoProtein. They detailed differences in how knowledge is integrated into language models, pre - trained, and new proteins are encoded. For instance, Kara directly uses knowledge as part of the input, while KeAP and OntoProtein implicitly embed knowledge within model parameters. However, authors should revise this paper and show that why the proposed method is quite distinct from previous work in both intuitive and empirical ways, and explicitly verify why directly uses knowledge as part of the input works better?

---

### Decision · Program_Chairs · 2025-01-22

Reject